# Subspace Identification for Multi-Source Domain Adaptation

**Zijian Li**[2,3], **Ruichu Cai**[2,*] **Guangyi Chen**[3,1], **Boyang Sun**[3], **Zhifeng Hao**[4], **Kun Zhang**[3,1*]

[1] Carnegie Mellon University
[2] School of Computer Science, Guangdong University of Technology
[3] Mohamed bin Zayed University of Artificial Intelligence
[4] Shantou University

## Abstract

Multi-source domain adaptation (MSDA) methods aim to transfer knowledge from multiple labeled source domains to an unlabeled target domain. Although current methods achieve target joint distribution identifiability by enforcing minimal changes across domains, they often necessitate stringent conditions, such as an adequate number of domains, monotonic transformation of latent variables, and invariant label distributions. These requirements are challenging to satisfy in real-world applications. To mitigate the need for these strict assumptions, we propose a subspace identification theory that guarantees the disentanglement of domain-invariant and domain-specific variables under less restrictive constraints regarding domain numbers and transformation properties, thereby facilitating domain adaptation by minimizing the impact of domain shifts on invariant variables. Based on this theory, we develop a Subspace Identification Guarantee (SIG) model that leverages variational inference. Furthermore, the SIG model incorporates class-aware conditional alignment to accommodate target shifts where label distributions change with the domains. Experimental results demonstrate that our SIG model outperforms existing MSDA techniques on various benchmark datasets, highlighting its effectiveness in real-world applications.

## 1 Introduction

Multi-Source Domain Adaptation (MSDA) is a method of transferring knowledge from multiple labeled source domains to an unlabeled target domain, to address the challenge of domain shift between the training data and the test environment. Mathematically, in the context of MSDA, we assume the existence of $M$ source domains $\{\mathcal{S}_1, \mathcal{S}_2, ..., \mathcal{S}_M\}$ and a single target domain $\mathcal{T}$. For each source domain $\mathcal{S}_i$, data are drawn from a distinct distribution, represented as $p_{\mathbf{x},\mathbf{y}|\mathbf{u}_{\mathcal{S}_i}}$, where the variables $\mathbf{x}, \mathbf{y}, \mathbf{u}$ correspond to features, labels, and domain indices, respectively. In a similar manner, the distribution within the target domain $\mathcal{T}$ is given by $p_{\mathbf{x},\mathbf{y}|\mathbf{u}_{\mathcal{T}}}$. In the source domains, we have access to $m_i$ annotated feature-label pairs of each domain, denoted by $(\mathbf{x}^{\mathcal{S}_i}, \mathbf{y}^{\mathcal{S}_i}) = (\mathbf{x}_k^{\mathcal{S}_i}, y_k^{Si})_{k=1}^{m_i}$, while in the target domain, only $m_{\mathcal{T}}$ unannotated features are observed, represented as $(\mathbf{x}^{(\mathcal{T})}) = (\mathbf{x}_k^{\mathcal{T}})_{k=1}^{m_{\mathcal{T}}}$. The primary goal of MSDA is to effectively leverage these labeled source data and unlabeled target data to identify the target joint distribution $p_{\mathbf{x},\mathbf{y}|\mathbf{u}_{\mathcal{T}}}$.

However, identifying the target joint distribution of $\mathbf{x}, \mathbf{y}|\mathbf{u}_{\mathcal{T}}$ using only $\mathbf{x}|\mathbf{u}_{\mathcal{T}}$ as observations present a significant challenge, since the possible mappings from $p_{\mathbf{x},\mathbf{y}|\mathbf{u}_{\mathcal{T}}}$ to $p_{\mathbf{x}|\mathbf{u}_{\mathcal{T}}}$ are infinite when no extra constraints are given. To solve this problem, some assumptions have been proposed to constrain the domain shift, such as covariate shift [44], target shift [58, 1], and conditional shift [3, 56]. For example, the most conventional covariate shift assumption posits that $p_{\mathbf{y}|\mathbf{x}}$ is fixed across different domains

---

*Corresponding authors.

37th Conference on Neural Information Processing Systems (NeurIPS 2023).

while $p_\mathbf{x}$ varies. Under this assumption, researchers can employ techniques such as importance reweighting [52], invariant representation learning [11], or cycle consistency [17] for distribution alignment. Additionally, target shift assumes that $p_{\mathbf{x}|\mathbf{y}}$ is fixed while the label distribution $p_\mathbf{y}$ changes, whereas conditional shift is represented by a fixed $p_\mathbf{y}$ and a varying $p_{\mathbf{x}|\mathbf{y}}$. More generally, the minimal change principle has been proposed, which not only unifies the aforementioned assumptions but also enables the theoretical guarantee of the identifiability of the target joint distribution. Specifically, it assumes that $p_{\mathbf{y}|\mathbf{u}}$ and $p_{\mathbf{x}|\mathbf{y},\mathbf{u}}$ change independently and the change of $p_{\mathbf{x}|\mathbf{y},\mathbf{u}}$ is minimal. Please refer to Appendix G for further discussion of related works, including domain adaptation and identification.

Although current methods demonstrate the identifiability of the target joint distribution through the minimal change principle, they often impose strict conditions on the data generation process and the number of domains, limiting their practical applicability. For instance, iMSDA [30] presents the component-wise identification of the domain-changed latent variables, subsequently identifying target joint distribution by modeling a data generation process with variational inference. However, this identification requires the following conditions. First, a sufficient number of auxiliary variables is employed for the component-wise theoretical guarantees, meaning that when the dimension of latent variables is $n$, a total of $2n + 1$ domains are needed. Second, in order to identify high-level invariant latent variables, a component-wise monotonic function between latent variables must be assumed. Third, these methods implicitly assume that label distribution remains stable across domains, despite the prevalence of target shift in real-world scenarios. These conditions are often too restrictive to be met in practice, highlighting the need for a more general approach to identifying latent variables across a wider range of domain shifts.

In an effort to alleviate the need for such strict assumptions, we present a subspace identification theory in this paper that guarantees the disentanglement of domain-invariant and domain-specific variables under more relaxed constraints concerning the number of domains and transformation properties. In contrast to component-wise identification, our subspace identification method demands fewer auxiliary variables (i.e., when the dimension of latent variables is $n$, only $n + 1$ domains are required). Additionally, we design a more general data generation process that accounts for target shift and does not necessitate monotonic transformation between latent variables. In this process, we categorize latent variables into four groups based on whether they are influenced by domain index or label. Building on the theory and causal generation process, we develop a Subspace Identification Guarantee (SIG) model that employs variational inference to identify latent variables. A class-aware condition alignment is incorporated to mitigate the impact of target shift, ensuring the update of the most confident cluster embedding. Our approach is validated through a simulation experiment for subspace identification evaluation and four widely-used public domain adaptation benchmarks for application evaluation. The impressive performance that outperforms state-of-the-art methods demonstrates the effectiveness of our method.

## 2 Identifying Target Joint Distribution with Data Generation Process

### 2.1 Data Generation Process

We begin with introducing the data generation process. As shown in Figure 1, the observed data $x \in \mathcal{X}$ are generated by latent variables $\mathbf{z} \in \mathcal{Z} \subseteq \mathbb{R}^n$. Sequentially, we divide the latent variables $\mathbf{z}$ into the four parts, i.e. $\mathbf{z} = \{\mathbf{z}_1, \mathbf{z}_2, \mathbf{z}_3, \mathbf{z}_4\} \in \mathcal{Z} \subseteq \mathbb{R}^n$, which are shown as follows.

- domain-specific and label-irrelevant variables $\mathbf{z}_1 \in \mathbb{R}^{n_1}$.
- domain-specific but label-relevant variables $\mathbf{z}_2 \in \mathbb{R}^{n_2}$.
- domain-invariant and label-relevant variables $\mathbf{z}_3 \in \mathbb{R}^{n_3}$.
- domain-invariant but label-irrelevant variables $\mathbf{z}_4 \in \mathbb{R}^{n_4}$.

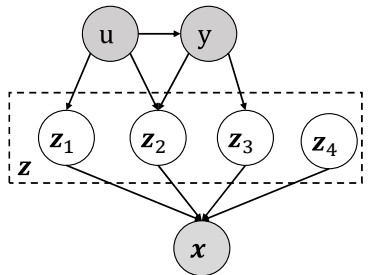

Figure 1: Data generation process, where the gray the white nodes denote the observed and latent variables, respectively.

To better understand these latent variables, we provide some examples in DomainNet datasets. First, $\mathbf{z}_1 \in \mathbb{R}^{n_1}$ denotes the styles of the images like "infograph" and "sketch", which are irrelevant to labels. $\mathbf{z}_2 \in \mathbb{R}^{n_2}$ denotes the latent variables that can be the texture information relevant to domains and labels. For example, the samples of "clock" and "telephone" contain some digits, and these digits in these samples are a special texture, which can be used for classification and be influenced by different

styles, such as "infograph". $\mathbf{z}_3 \in \mathbb{R}^{n_3}$ denotes the latent variables that are only relevant to the labels. For example, in the DomainNet dataset, it can be interpreted as the meaning of different classes like "Bicycle" or "Teapot". Finally, $\mathbf{z}_4 \in \mathbb{R}^{n_4}$ denotes the label-irrelevant latent variables. For example, $\mathbf{z}_4$ can be interpreted as the background that is invariant to domains and labels.

Based on the definitions of these latent variables, we let the observed data be generated from $\mathbf{z}$ through an invertible and smooth mixing function $g : \mathcal{Z} \rightarrow \mathcal{X}$. Due to the target shift, we further consider that the $p_{\mathbf{y}}$ is influenced by $\mathbf{u}$, i.e. $\mathbf{u} \rightarrow \mathbf{y}$.

Compared with the existing data generation process like [30], the proposed data generation process is different in three folds. First, $p_{\mathbf{u}}$ is independent of $p_{\mathbf{y}}$ in the iMSDA [30], so the target shift is not taken into account. Second, the data generation process of iMSDA requires an invertible and monotonic function between latent variables for component-wise identification, which is too strict to be met in practice. Third, to provide a more general way to depict the real-world data, our data generation process introduces the domain-specific but label-relevant latent variables $\mathbf{z}_{s_2}$ and the domain-invariant but label-irrelevant variables $\mathbf{z}_4$.

## 2.2 Identifying the Target Joint Distribution

In this part, we show how to identify the target joint distribution $p_{\mathbf{x},\mathbf{y}|\mathbf{u}_{\mathcal{T}}}$ with the help of marginal distribution. By introducing the latent variables and combining the proposed data generation process, we can obtain the following derivation.

$$
\begin{aligned}
p_{\mathbf{x},\mathbf{y}|\mathbf{u}_{\mathcal{T}}} &= \int_{\mathbf{z}_1} \int_{\mathbf{z}_2} \int_{\mathbf{z}_3} \int_{\mathbf{z}_4} p_{\mathbf{x},\mathbf{y},\mathbf{z}_1,\mathbf{z}_2,\mathbf{z}_3,\mathbf{z}_4|\mathbf{u}_{\mathcal{T}}} d\mathbf{z}_1 d\mathbf{z}_2 d\mathbf{z}_3 d\mathbf{z}_4 \\
&= \int_{\mathbf{z}_1} \int_{\mathbf{z}_2} \int_{\mathbf{z}_3} \int_{\mathbf{z}_4} p_{\mathbf{x},\mathbf{z}_1,\mathbf{z}_2,\mathbf{z}_3,\mathbf{z}_4|\mathbf{y},\mathbf{u}_{\mathcal{T}}} \cdot p_{\mathbf{y}|\mathbf{u}_{\mathcal{T}}} d\mathbf{z}_1 d\mathbf{z}_2 d\mathbf{z}_3 d\mathbf{z}_4 \\
&= \int_{\mathbf{z}_1} \int_{\mathbf{z}_2} \int_{\mathbf{z}_3} \int_{\mathbf{z}_4} p_{\mathbf{x}|\mathbf{z}_1,\mathbf{z}_2,\mathbf{z}_3,\mathbf{z}_4} \cdot p_{\mathbf{z}_1,\mathbf{z}_2,\mathbf{z}_3,\mathbf{z}_4|\mathbf{y},\mathbf{u}_{\mathcal{T}}} \cdot p_{\mathbf{y}|\mathbf{u}_{\mathcal{T}}} d\mathbf{z}_1 d\mathbf{z}_2 d\mathbf{z}_3 d\mathbf{z}_4.
\end{aligned}
\tag{1}
$$

According to the derivation in Equation (1), we can identify the target joint distribution by modeling three distributions. First, we need to model $p_{\mathbf{x}|\mathbf{z}_1,\mathbf{z}_2,\mathbf{z}_3,\mathbf{z}_4}$, implying that we need to model the conditional distribution of observed data give latent variables, which coincides with a generative model for observed data. Second, we need to estimate the label pseudo distribution of target domain $p_{\mathbf{y}|\mathbf{u}_{\mathcal{T}}}$. Third, we need to model $p_{\mathbf{z}_1,\mathbf{z}_2,\mathbf{z}_3,\mathbf{z}_4|\mathbf{y},\mathbf{u}_{\mathcal{T}}}$ meaning that the latent variables should be identified with auxiliary variables $\mathbf{u}, \mathbf{y}$ under theoretical guarantees. In the next section, we will introduce how to identify these latent variables with subspace identification block-wise identification results.

## 3 Subspace Identifiability for Latent Variables

In this section, we provide how to identify the latent variables in Figure 1. In detail, we first prove that $\mathbf{z}_2$ is subspace identifiable and $\mathbf{z}_1, \mathbf{z}_3$ can be reconstructed from the estimated $\hat{\mathbf{z}}_1, \hat{\mathbf{z}}_2, \hat{\mathbf{z}}_3$. Then we further prove that $\mathbf{z}_4$ is block-wise identifiable.

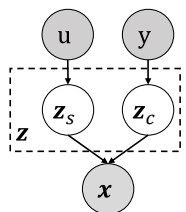

To clearly introduce the subspace identification theory, we employ a simple data generation process [3] as shown in Figure 2. In this data generation process, $\mathbf{z}_s \in \mathbb{R}^{n_s}$ and $\mathbf{z}_c \in \mathbb{R}^{n_c}$ denote the domain-specific and domain-invariant latent variables, respectively. For convenient, we let $\mathbf{z} = \{\mathbf{z}_s, \mathbf{z}_c\}, n = n_s + n_c$. Moreover, we assume $\mathbf{z}_s = (z_i)_{i=1}^{n_s}$ and $\mathbf{z}_c = (z_i)_{i=n_s+1}^{n}$. And $\{\mathbf{u}, \mathbf{y}, \mathbf{x}\}$ denote the domain index, labels, and observed data, respectively. And we further let the observed data be generated from $\mathbf{z}$ through an invertible and smooth mixing function $g : \mathcal{Z} \rightarrow \mathcal{X}$. The subspace identification of $\mathbf{z}_s$ means that for each ground-truth $z_{s,i}$, there exits $\hat{\mathbf{z}}_s$ and an invertible function $h_i : \mathbb{R}^n \rightarrow \mathbb{R}$, such that $z_{s,i} = h_i(\hat{\mathbf{z}}_s)$.

Figure 2: A simple data generalization process for introducing subspace identification.

**Theorem 1.** (*Subspace Identification of* $\mathbf{z}_s$.) *We follow the data generation process in Figure 2 and make the following assumptions:*

- *A1 (Smooth and Positive Density): The probability density function of latent variables is smooth and positive, i.e., $p_{\mathbf{z}|\mathbf{u}} > 0$ over $\mathcal{Z}$ and $\mathcal{U}$.*

- A2 *(Conditional independent)*: *Conditioned on* $\mathbf{u}$*, each* $z_i$ *is independent of any other* $z_j$ *for* $i, j \in \{1, \cdots, n\}, i \neq j$*, i.e.* $\log p_{\mathbf{z}|\mathbf{u}}(\mathbf{z}|\mathbf{u}) = \sum_i^n q_i(z_i, \mathbf{u})$ *where* $q_i(z_i, \mathbf{u})$ *is the log density of the conditional distribution, i.e.,* $q_i : \log p_{z_i|\mathbf{u}}$.
- A3 *(Linear independence)*: *For any* $\mathbf{z}_s \in \mathcal{Z}_s \subseteq \mathbb{R}^{n_s}$*, there exist* $n_s + 1$ *values of* $\mathbf{u}$*, i.e.,* $\mathbf{u}_j$ *with* $j = 0, 1, \cdots, n_s$*, such that these* $n_s$ *vectors* $\mathbf{w}(\mathbf{z}, \mathbf{u}_j) - \mathbf{w}(\mathbf{z}, \mathbf{u}_0)$ *with* $j = 1, \cdots, n_s$ *are linearly independent, where vector* $\mathbf{w}(\mathbf{z}, \mathbf{u}_j)$ *is defined as follows:*

$$\mathbf{w}(\mathbf{z}, \mathbf{u}) = \left( \frac{\partial q_1(z_1, \mathbf{u})}{\partial z_1}, \cdots, \frac{\partial q_i(z_i, \mathbf{u})}{\partial z_i}, \cdots \frac{\partial q_{n_s}(z_{n_s}, \mathbf{u})}{\partial z_{n_s}} \right), \tag{2}$$

*By modeling the aforementioned data generation process,* $\mathbf{z}_s$ *is subspace identifiable.*

**Proof sketch.** First, we construct an invertible transformation $h$ between the ground-truth $\mathbf{z}$ and estimated $\hat{\mathbf{z}}$. Sequentially, we leverage the variance of different domains to construct a full-rank linear system, where the only solution of $\frac{\partial \mathbf{z}_s}{\partial \hat{\mathbf{z}}_c}$ is zero. Since the Jacobian of $h$ is invertible, for each $z_{s,i}, i \in \{1, \cdots, n_s\}$, there exists a $h_i$ such that $z_{s,i} = h_i(\hat{\mathbf{z}})$ and $\mathbf{z}_s$ is subspace identifiable.

The proof can be found in Appendix B.1. The first two assumptions are standard in the component-wise identification of existing nonlinear ICA [30, 27]. The third Assumption means that $p_{\mathbf{z}|\mathbf{u}}$ should vary sufficiently over $n+1$ domains. Compared to component-wise identification, which necessitates $2n+1$ domains and is likely challenging to fulfill, subspace identification can yield equivalent results in terms of identifying the ground-truth latent variables with only $n+1$ domains. Therefore, subspace identification benefits from a more relaxed assumption.

Based on the theoretical results of subspace identification, we show that the ground-truth $\mathbf{z}_1, \mathbf{z}_2$ and $\mathbf{z}_3$ be reconstructed from the estimated $\hat{\mathbf{z}}_1, \hat{\mathbf{z}}_2$ and $\hat{\mathbf{z}}_3$. For ease of exposition, we assume that $\mathbf{z}_1, \mathbf{z}_2, \mathbf{z}_3$, and $\mathbf{z}_4$ correspond to components in $\mathbf{z}$ with indices $\{1, \cdots, n_1\}, \{n_1 + 1, \cdots, n_1 + n_2\}, \{n_1 + n_2 + 1, \cdots, n_1 + n_2 + n_3\}$, and $\{n_1 + n_2 + n_3 + 1, \cdots, n\}$, respectively.

**Corollary 1.1.** *We follow the data generation in Section 2.1, and make the following assumptions which are similar to A1-A3:*

*A4 (Smooth and Positive Density): The probability density function of latent variables is smooth and positive, i.e.,* $p_{\mathbf{z}|\mathbf{u},\mathbf{y}} > 0$ *over* $\bar{\mathcal{Z}}, \mathcal{U},$ *and* $\mathcal{Y}$.

*A5 (Conditional independent): Conditioned on* $\mathbf{u}$ *and* $\mathbf{y}$*, each* $z_i$ *is independent of any other* $z_j$ *for* $i, j \in \{1, \cdots, n\}, i \neq j$*, i.e.* $\log p_{\mathbf{z}|\mathbf{u},\mathbf{y}}(\mathbf{z}|\mathbf{u}, \mathbf{y}) = \sum_i^n q_i(z_i, \mathbf{u}, \mathbf{y})$ *where* $q_i(z_i, \mathbf{u}, \mathbf{y})$ *is the log density of the conditional distribution, i.e.,* $q_i : \log p_{z_i|\mathbf{u},\mathbf{y}}$.

*A6 (Linear independence): For any* $\mathbf{z} \in \mathcal{Z} \subseteq \mathbb{R}^n$*, there exists* $n_1 + n_2 + n_3 + 1$ *combination of* $(\mathbf{u}, \mathbf{y})$*, i.e.* $j = 1, \cdots, U$ *and* $c = 1, \cdots, C$ *and* $U \times C - 1 = n_1 + n_2 + n_3$*, where* $U$ *and* $C$ *denote the number of domains and the number of labels. such that these* $n_1 + n_2 + n_3$ *vectors* $\mathbf{w}(\mathbf{z}, \mathbf{u}_j, \mathbf{y}_c) - \mathbf{w}(\mathbf{z}, \mathbf{u}_0, \mathbf{y}_0)$ *are linearly independent, where* $\mathbf{w}(\mathbf{z}, \mathbf{u}_j, \mathbf{y}_c)$ *is defined as follows:*

$$\mathbf{w}(\mathbf{z}, \mathbf{u}, \mathbf{y}) = \left( \frac{\partial q_1(z_1, \mathbf{u}, \mathbf{y})}{\partial z_1}, \cdots, \frac{\partial q_i(z_i, \mathbf{u}, \mathbf{y})}{\partial z_i}, \cdots \frac{\partial q_n(z_n, \mathbf{u}, \mathbf{y})}{\partial z_n} \right). \tag{3}$$

*By modeling the data generation process in Section 2.1,* $\mathbf{z}_2$ *is subspace identifiable, and* $\mathbf{z}_1, \mathbf{z}_3$ *can be reconstructed from* $\hat{\mathbf{z}}_1, \hat{\mathbf{z}}_2$ *and* $\hat{\mathbf{z}}_2, \hat{\mathbf{z}}_3$*, respectively.*

**Proof sketch.** The detailed proof can be found in Appendix B.2. First, we construct an invertible transformation $h$ to bridge the relation between the ground-truth $\mathbf{z}$ and the estimated $\hat{\mathbf{z}}$. Then, we repeatedly use Theorem 1 three times by considering the changing of labels and domains. Hence, we find that the values of some blocks of the Jacobian of $h$ are zero. Finally, the Jacobian of $h$ can be formalized as Equation (4). where $\boldsymbol{J}_h$ denotes the Jacobian of $h$ and $\boldsymbol{J}_h^{ij} := \frac{\partial \mathbf{z}_i}{\partial \hat{\mathbf{z}}_j}$ and $i, j \in \{1, 2, 3, 4\}$. Since $h(\cdot)$ is invertible, $\boldsymbol{J}_h$ is a full-rank matrix. Therefore, for each $z_{2,i}, i \in \{n_1 + 1, \cdots, n_1 + n_2\}$, there exists a $h_{2,i}$ such that $z_{2,i} = h_i(\hat{\mathbf{z}}_2)$. Moreover, for each $z_{1,i}, i \in \{1, \cdots, n_1 + 1\}$, there exists a

$$\boldsymbol{J}_h = \begin{bmatrix} \boldsymbol{J}_h^{1,1} & \boldsymbol{J}_h^{1,2} & \boldsymbol{J}_h^{1,3} = 0 & \boldsymbol{J}_h^{1,4} = 0 \\ \boldsymbol{J}_h^{2,1} = 0 & \boldsymbol{J}_h^{2,2} & \boldsymbol{J}_h^{2,3} = 0 & \boldsymbol{J}_h^{2,4} = 0 \\ \boldsymbol{J}_h^{3,1} = 0 & \boldsymbol{J}_h^{3,2} & \boldsymbol{J}_h^{3,3} & \boldsymbol{J}_h^{3,4} = 0 \\ \boldsymbol{J}_h^{4,1} & \boldsymbol{J}_h^{4,2} & \boldsymbol{J}_h^{4,3} & \boldsymbol{J}_h^{4,4} \end{bmatrix}, \tag{4}$$

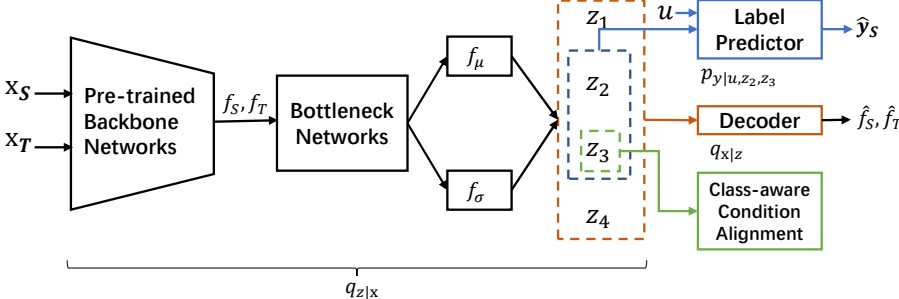

Figure 3: The framework of the Subspace Identification Guarantee model. The pre-trained backbone networks are used to extract the feature $f$ from observed data. The bottleneck and $f_\mu, f_\sigma$ are used to generate $\mathbf{z}$ with a reparameterization trick. Label predictor takes $\mathbf{z}_2, \mathbf{z}_3$, and $\mathbf{u}$ as input to model $p_{\mathbf{y}|\mathbf{u},\mathbf{z}_2,\mathbf{z}_3}$. The decoder is used to model the marginal distribution. Finally, $\mathbf{z}_2$ is used for class-aware conditional alignment.

$h_{1,i}$ such that $z_{1,i} = h_{1,i}(\hat{\mathbf{z}}_1, \hat{\mathbf{z}}_2)$. And for each $z_{3,i}, i \in \{n_1 + n_2 + 1, \cdots, n_1 + n_2 + n_3\}$, there exists a $h_{3,i}$ such that $z_{3,i} = h_{3,i}(\hat{\mathbf{z}}_2, \hat{\mathbf{z}}_3)$. Then we prove that $\mathbf{z}_4$ is block-wise identifiable, which means that there exists an invertible function $h_4, s.t.\mathbf{z}_4 = h_4(\hat{\mathbf{z}}_4)$.

**Lemma 2.** *[30] Following the data generation process in Section 2.1 and the assumptions A4-A6 in Theorem 3, we further make the following assumption:*

- *A7 (Domain Variability: For any set $A_{\mathbf{z}} \subseteq \mathcal{Z}$) with the following two properties: 1) $A_{\mathbf{z}}$ has nonzero probability measure, i.e. $\mathbb{P}[\mathbf{z} \in A_{\mathbf{z}}|\{\mathbf{u} = \mathbf{u}', \mathbf{y} = \mathbf{y}'\}] > 0$ for any $\mathbf{u}' \in \mathcal{U}$ and $\mathbf{y}' \in \mathcal{Y}$. 2) $A_{\mathbf{z}}$ cannot be expressed as $B_{\mathbf{z}_4} \times \mathcal{Z}_1 \times \mathcal{Z}_2 \times \mathcal{Z}_3$ for any $B_{\mathbf{z}_4} \subset \mathcal{Z}_4$.*

$\exists \mathbf{u}_1, \mathbf{u}_2 \in \mathcal{U}$ and $\mathbf{y}_1, \mathbf{y}_2 \in \mathcal{Y}$, such that $\int_{\mathbf{z} \in \mathcal{A}_{\mathbf{z}}} p_{\mathbf{z}|\mathbf{u}_1, \mathbf{y}_1} d\mathbf{z} \neq \int_{\mathbf{z} \in \mathcal{A}_{\mathbf{z}}} p_{\mathbf{z}|\mathbf{u}_2, \mathbf{y}_2} d\mathbf{z}$. *By modeling the data generation process in Section 2.1, the $\mathbf{z}_4$ is block-wise identifiable.*

The proof can be found in Appendix B.3. Lemma 4 shows that $\mathbf{z}_4$ can be block-wise identifiable when the $p_{\mathbf{z}|\mathbf{u}}$ changes sufficiently across domains.

In summary, we can obtain the estimated latent variables $\hat{\mathbf{z}}$ with the help of subspace identification and block-wise identification.

## 4 Subspace Identification Guarantee Model

Based on the theoretical results, we proposed the Subspace Identification Guaranteed model (SIG) as shown in Figure 3, which contains a variational-inference-based neural architecture to model the marginal distribution and a class-aware conditional alignment to mitigate the target shift.

### 4.1 Variational-Inference-based Neural Architecture

According to the data generation process in Figure 1, we first derive the evidence lower bound (ELBO) in Equation (5).

$$ELBO = \mathbb{E}_{q_{\mathbf{z}|\mathbf{x}}(\mathbf{z}|\mathbf{x})} \ln p_{\mathbf{x}|\mathbf{z}}(\mathbf{x}|\mathbf{z}) + \mathbb{E}_{q_{\mathbf{z}|\mathbf{x}}(\mathbf{z}|\mathbf{x})} \ln p_{\mathbf{y}|\mathbf{u},\mathbf{z}_2,\mathbf{z}_3}(\mathbf{y}|\mathbf{u}, \mathbf{z}_2, \mathbf{z}_3)$$
$$+ \mathbb{E}_{q_{\mathbf{z}|\mathbf{x}}(\mathbf{z}|\mathbf{x})} \ln p_{\mathbf{u}|\mathbf{z}}(\mathbf{u}|\mathbf{z}) - D_{KL}(q_{\mathbf{z}|\mathbf{x}}(\mathbf{z}|\mathbf{x})||p_{\mathbf{z}}(\mathbf{z})). \tag{5}$$

Since the reconstruction of $\mathbf{u}$ is not the optimization goal, we remove the reconstruction of $\mathbf{u}$ and we rewrite Equation (5) as the objective function in Equation (6).

$$\mathcal{L}_{elbo} = \mathcal{L}_{vae} + \mathcal{L}_y$$
$$\mathcal{L}_{vae} = -\mathbb{E}_{q_{\mathbf{z}|\mathbf{x}}(\mathbf{z}|\mathbf{x})} \ln p_{\mathbf{x}|\mathbf{z}}(\mathbf{x}|\mathbf{z}) + D_{KL}(q_{\mathbf{z}|\mathbf{x}}(\mathbf{z}|\mathbf{x})||p_{\mathbf{z}}(\mathbf{z})) \tag{6}$$
$$\mathcal{L}_y = -\mathbb{E}_{q_{\mathbf{z}|\mathbf{x}}(\mathbf{z}|\mathbf{x})} \ln p_{\mathbf{y}|\mathbf{u},\mathbf{z}_2,\mathbf{z}_3}(\mathbf{y}|\mathbf{u}, \mathbf{z}_2, \mathbf{z}_3).$$

To minimize the pairwise class confusion, we further employ the minimum class confusion [25] into the classification loss $\mathcal{L}_y$. According to the objective function in Equation (6), we illustrate how to

implement the SIG model in Figure 3. First, we take the observed data $\mathbf{x}_{S_i}$ and $\mathbf{x}_T$ from the source domains and the target domain as the inputs of the pre-trained backbone networks like ResNet50 and extract the feature $f_{S_i}$ and $f_T$. Sequentially, we employ an MLP-based encoder, which contains bottleneck networks, $f_\mu$ and $f_\sigma$, to extract the latent variables $\mathbf{z}$. Then, we take $\mathbf{z}$ to reconstruct the pre-trained features via an MLP-based decoder to estimate the marginal distribution $p_{\mathbf{x}|\mathbf{u}}$. Finally, we take the $\mathbf{z}_2, \mathbf{z}_3$, and the domain embedding to predict the source label to estimate $p_{\mathbf{y}|\mathbf{u},\mathbf{z}_2,\mathbf{z}_3}$.

### 4.2 Class-aware Conditional Alignment

To estimate the target label distribution $p_{\mathbf{y}|\mathbf{u}_T}$ and mitigate the influence of target shift, we propose the class-aware conditional alignment to automatically adjust the conditional alignment loss for each sample. Formally, it can be written as

$$\mathcal{L}_a = \frac{1}{C}\sum_{i=1}^{C} w^{(i)} \cdot p_{\hat{\mathbf{y}}^{(i)}} ||\hat{\mathbf{z}}_{3,\mathcal{S}}^{(i)} - \hat{\mathbf{z}}_{3,\mathcal{T}}^{(i)}||_2, \quad w^{(i)} = 1 + \exp\left(-H(p_{\hat{\mathbf{y}}^{(i)}})\right), \tag{7}$$

where $C$ denotes the class number; $\hat{\mathbf{z}}_{3,\mathcal{S}}^{(i)}$ and $\hat{\mathbf{z}}_{3,\mathcal{T}}^{(i)}$ denote the latent variables of $i$th class from source and target domain, respectively; $w^{(i)}$ denotes the prediction uncertainty of each class in the target domain; $p_{\hat{\mathbf{y}}^{(i)}}$ denotes the estimated label probability density of $i$th class; $H$ denotes the entropy.

The aforementioned class-aware conditional alignment is based on the existing conditional alignment loss, which can be formalized in Equation (8).

$$\mathcal{L}_a = \frac{1}{C}\sum_{i=1}^{C} ||\hat{\mathbf{z}}_{3,\mathcal{S}}^{(i)} - \hat{\mathbf{z}}_{3,\mathcal{T}}^{(i)}||_2, \tag{8}$$

However, conventional conditional alignment usually suffers from two drawbacks including misestimated centroid and low-quality pseudo-labels. First, the conditional alignment method implicitly assumes that the feature centroids from different domains are the same. But it is hard to estimate the correct centroids of the target domain for the class with low probability density. Second, conditional alignment heavily relies on the quality of the pseudo label. But existing methods usually use pseudo labels without any discrimination, which might result in false alignment. To solve these problems, we consider two types of reweighting.

**Distribution-based Reweighting for Misestimated Centroid:** Although the conditional alignment method implicitly assumes that the feature centroids from different domains are the same, it is hard to estimate the correct centroids of the target domain for the class with low probability density. To address this challenge, one straightforward solution is to consider the label distribution of the target domain. To achieve this, we employ the technique of black box shift estimation method (BBSE) [37] to estimate the label distribution from the target domain $p_{\hat{\mathbf{y}}}$. So we use the estimated label distribution to reweight the conditional alignment loss in Equation (8).

**Entropy-based Reweighting for Low-quality Pseudo-labels:** conditional alignment heavily relies on the quality of the pseudo label. However, existing methods usually use pseudo labels without any discrimination, which might result in false alignment. To address this challenge, we consider the prediction uncertainty of each class in the target domain. Technologically, for each sample in the target dataset, we calculate the entropy-based weights via the prediction results which are shown as $w^{(i)}$ in Equation (7).

By combining the distribution-based weights and the entropy-based weight, we can obtain the class-aware conditional alignment as shown in Equation (7). Hence the total loss of the Subspace Identification Guarantee model can be formalized as follows:

$$\mathcal{L}_{total} = \mathcal{L}_y + \beta\mathcal{L}_{vae} + \alpha\mathcal{L}_a, \tag{9}$$

where $\alpha, \beta$ denote the hyper-parameters.

## 5 Experiments

### 5.1 Experiments on Simulation Data

In this subsection, we illustrate the experiment results of simulation data to evaluate the theoretical results of subspace identification in practice.

### 5.1.1 Experimental Setup

**Data Generation.** We generate the simulation data for binary classification with 8 domains. To better evaluate our theoretical results, we follow the data generation process in Figure 2, which includes two types of latent variables, i.e., domain-specific latent variables $\mathbf{z}_s$ and domain-invariant latent variables $\mathbf{z}_c$. We let the dimensions of $\mathbf{z}_s$ and $\mathbf{z}_c$ be both 2. Moreover, $\mathbf{z}_s$ are sampled from $u$ different mixture of Gaussians, and $\mathbf{z}_c$ are sampled from a factorized Gaussian distribution. We let the data generation process from latent variables to observed variables be MLPs with the Tanh activation function. We further split the simulation dataset into the training set, validation set, and test set.

**Evaluation Metrics.** First, we employ the accuracy of the target domain data to measure the classification performance of the model. Second, we compute the Mean Correlation Coefficient (MCC) between the ground-truth $\mathbf{z}_s$ and the estimated $\hat{\mathbf{z}}_s$ on the test dataset to evaluate the component-wise identifiability of domain-specific latent variables. A higher MCC denotes the better identification performance the model can achieve. Third, to evaluate the performance of subspace identifiability of domain-specific latent variables, we first use the estimated $\hat{\mathbf{z}}_s$ from the validation set to regress each dimension of the ground-truth $\mathbf{z}_s$ from the validation set with the help of a MLPs. Sequentially, we take the $\hat{\mathbf{z}}_s$ from the test set as input to estimate how well the MLPs can reconstruct the ground-truth $\mathbf{z}_s$ from the test set, so we employ Root Mean Square Error (RMSE) to measure the extent of subspace identification. A low RMSE denotes that there exists a transformation $h_i$ between $\mathbf{z}_{s,i}$ and $\hat{\mathbf{z}}_{s,1}, \hat{\mathbf{z}}_{s,2}$, i.e. $\mathbf{z}_{s,i} = h_i(\hat{\mathbf{z}}_{s,1}, \hat{\mathbf{z}}_{s,2}), i \in \{1, 2\}$. For the scenario where the number of domains is less than 8, we first fix the target domain and then try all the combinations of the source domains. And we publish the average performance of all the combinations. We repeat each experiment over 3 random seeds.

### 5.1.2 Results and Discussion

The experimental results of the simulation dataset are shown in Table 1. According to the experiment results, we can obtain the following conclusions:
1) We can find

Table 1: Experiments results on simulation data.

| State | U | ACC | MCC | RMSE |
|---|---|---|---|---|
| Component-wise Identification | 8 | 0.9982(0.0004) | 0.9037(0.0087) | 0.0433(0.0051) |
| | 6 | 0.9982(0.0007) | 0.8976(0.0162) | 0.0439(0.0073) |
| | 5 | 0.9982(0.0007) | 0.8973(0.0131) | 0.0441(0.0055) |
| Subspace Identification | 4 | 0.9233(0.2039) | 0.8484(0.1452) | 0.0582(0.0431) |
| | 3 | 0.8679(0.2610) | 0.8077(0.1709) | 0.0669(0.0482) |
| No Identification | 2 | 0.5978(0.3039) | 0.6184(0.2093) | 0.1272(0.0608) |

that the values of MCC increase along with the number of domains. Moreover, the values of MCC are high (around 0.9) and stable when the number of domains is larger than 5. This result corresponds to the theoretical result of component-wise identification, where a certain number of domains (i.e. $2n + 1$) are necessary for component-wise identification. 2) We can find that the values of RMSE decrease along with the number of domains. Furthermore, the values of RMSE are low and stable (less than 0.07) when the number of domains is larger than 3, but it drops sharply when $u = 2$. These experimental results coincide with the theoretical results of subspace identification as well as the intuition where a certain number of domains are necessary for subspace identification (i.e. $n_s + 1$). 3) According to the experimental results of ACC, we can find that the accuracy grows along with the number of domains and its changing pattern is relevant to that of RMSE, i.e., the performance is stable when the number of domains is larger than 3. The ACC results also indirectly support the results of subspace identification, since one straightforward understanding of subspace identification is that the domain-specific information is preserved in $\hat{z}_s$. And the latent variables are well disentangled with the help of subspace identification, which benefits the model performance.

## 5.2 Experiments on Real-world Data

### 5.2.1 Experimental Setup

**Datasets:** We consider four benchmarks: Office-Home, PACS, ImageCLEF, and DomainNet. For each dataset, we let each domain be a target domain and the other domains be the source domains. For the DomainNet dataset, we equip a cross-attention module to the ResNet101 backbone networks for

Table 2: Classification results on the Office-Home and ImageCLEF datasets. For the Office-Home dataset, We employ ResNet50 as the backbone network. For the ImageCLEF dataset, we employ ResNet18 as the backbone network.

| Model | Office-Home | | | | | ImageCLEF | | | |
|---|---|---|---|---|---|---|---|---|---|
| | Art | Clipart | Product | RealWorld | Average | P | C | I | Average |
| **Source Only [16]** | 64.5 | 52.3 | 77.6 | 80.7 | 68.8 | 77.2 | 92.3 | 88.1 | 85.8 |
| **DANN [11]** | 64.2 | 58.0 | 76.4 | 78.8 | 69.3 | 77.9 | 93.7 | 91.8 | 87.8 |
| **DAN [40]** | 68.2 | 57.9 | 78.4 | 81.9 | 71.6 | 77.6 | 93.3 | 92.2 | 87.7 |
| **DCTN [69]** | 66.9 | 61.8 | 79.2 | 77.7 | 71.4 | 75.0 | 95.7 | 90.3 | 87.0 |
| **MFSAN [81]** | 72.1 | 62.0 | 80.3 | 81.8 | 74.1 | 79.1 | 95.4 | 93.6 | 89.4 |
| **WADN [54]** | 75.2 | 61.0 | 83.5 | 84.4 | 76.1 | 77.7 | 95.8 | 93.2 | 88.9 |
| **iMSDA [30]** | 75.4 | 61.4 | 83.5 | 84.4 | 76.2 | 79.2 | 96.3 | **94.3** | 90.0 |
| **SIG** | **76.4** | **63.9** | **85.4** | **85.8** | **77.8** | **79.3** | **97.3** | **94.3** | **90.3** |

better usage of domain knowledge. We also employ the alignment of MDD [78]. For the Office-Home and ImageCLEF datasets, we employ the pre-trained ResNet50 with an MLP-based classifier. For the PACS dataset, we use ResNet18 with an MLP-based classifier. The implementation details are provided in the Appendix C. We report the average results over 3 random seeds.

**Baselines:** Besides the classical approaches for single source domain adaptation like DANN [11], DAN [40], MCD [50], and ADDA [61]. We also compare our method with several state-of-the-art multi-source domain adaptation methods, for example, MIAN-$\gamma$ [45], T-SVDNet [33], LtC-MSDA [63], SPS [64], and PFDA [10]. Moreover, we further consider the WADN [54], which is devised for the target shift of multi-source domain adaptation. For a fair comparison, we employ the same pre-train backbone networks instead of the pre-trained features for WADN in the original paper. We also consider the latest iMSDA [30], which addresses the MSDA via component-wise identification.

Table 3: Classification results on the PACS datasets. We employ ResNet18 as the backbone network. Experiment results of other compared methods are taken from ([30]).

| Model | A | C | P | S | Average |
|---|---|---|---|---|---|
| **Source Only [16]** | 74.9 | 72.1 | 94.5 | 64.7 | 76.7 |
| **DANN [11]** | 81.9 | 77.5 | 91.8 | 74.6 | 81.5 |
| **MDAN [79]** | 79.1 | 76.0 | 91.4 | 72.0 | 79.6 |
| **WBN [43]** | 89.9 | 89.7 | 97.4 | 58.0 | 83.8 |
| **MCD [50]** | 88.7 | 88.9 | 96.4 | 73.9 | 87.0 |
| **M3SDA [46]** | 89.3 | 89.9 | 97.3 | 76.7 | 88.3 |
| **CMSS [70]** | 88.6 | 90.4 | 96.9 | 82.0 | 89.5 |
| **LtC-MSDA [63]** | 90.1 | 90.4 | 97.2 | 81.5 | 89.8 |
| **T-SVDNet [33]** | 90.4 | 90.6 | 98.5 | 85.4 | 91.2 |
| **iMSDA [30]** | 93.7 | 92.4 | 98.4 | 89.2 | 93.4 |
| **SIG** | **94.1** | **93.6** | **98.6** | **89.5** | **93.9** |

### 5.2.2 Results and Discussion

Experimental results on Office-Home, ImageCLEF, PACS, and DomainNet are shown in Table 2, 3, and 4, respectively. Experiment results of other compared methods are provided in Appendix D.2.

According to the experiment results of the Office-Home dataset on the left side of Table 2, our SIG model significantly outperforms all other baselines on all the transfer tasks. Specifically, our method outperforms the most competitive baseline by a clear margin of $1.3\% - 4\%$ and promotes the classification accuracy substantially on the hard transfer task, e.g. Clipart. It is noted that our method achieves a better performance than that of WADN, which is designed for the target shift scenario. This is because our method not only considers how the domain variables influence the distribution of labels but also identifies the latent variables of the data generation process. Moreover, our SIG method also outperforms iMSDA, indirectly reflecting that the proposed data generation process is closer to the real-world setting and the subspace identification can achieve better disentanglement performance under limited auxiliary variables.

For datasets like ImageCLEF and PACS, our method also achieves the best-averaged results. In detail, we achieved a comparable performance in all the transfer tasks in the ImageCLFE dataset. In the PACS dataset, our SIG method still performs better than the latest compared methods like iMSDA and T-SVDNet in some challenging transfer tasks like Cartoon. Finally, we also consider the

Table 4: Classification results on the DomainNet datasets. We employ ResNet101 as the backbone network. Experiment results of other compared methods are taken from ([34] and [64]).

| Model | Clipart | Infograph | Painting | Quickdraw | Real | Sketch | Average |
|---|---|---|---|---|---|---|---|
| **Source Only** [16] | 52.1 | 23.4 | 47.6 | 13.0 | 60.7 | 46.5 | 40.6 |
| **ADDA** [61] | 47.5 | 11.4 | 36.7 | 14.7 | 49.1 | 33.5 | 32.2 |
| **MCD** [50] | 54.3 | 22.1 | 45.7 | 7.6 | 58.4 | 43.5 | 38.5 |
| **DANN** [11] | 60.6 | 25.8 | 50.4 | 7.7 | 62.0 | 51.7 | 43 |
| **DCTN** [69] | 48.6 | 23.5 | 48.8 | 7.2 | 53.5 | 47.3 | 38.2 |
| **M$^3$SDA-$\beta$** [46] | 58.6 | 26.0 | 52.3 | 6.3 | 62.7 | 49.5 | 42.6 |
| **ML_MSDA** [35] | 61.4 | 26.2 | 51.9 | 19.1 | 57.0 | 50.3 | 44.3 |
| **meta-MCD** [32] | 62.8 | 21.4 | 50.5 | 15.5 | 64.6 | 50.4 | 44.2 |
| **LtC-MSDA** [63] | 63.1 | 28.7 | 56.1 | 16.3 | 66.1 | 53.8 | 47.4 |
| **CMSS** [70] | 64.2 | 28.0 | 53.6 | 16.9 | 63.4 | 53.8 | 46.5 |
| **DRT+ST** [34] | 71.0 | 31.6 | 61.0 | 12.3 | 71.4 | **60.7** | 51.3 |
| **SPS** [64] | 70.8 | 24.6 | 55.2 | 19.4 | 67.5 | 57.6 | 49.2 |
| **PFDA** [10] | 64.5 | 29.2 | 57.6 | 17.2 | 67.2 | 55.1 | 48.5 |
| **iMSDA** [30] | 68.1 | 25.9 | 57.4 | 17.3 | 64.2 | 52.0 | 47.5 |
| **SIG** | **72.7** | **32.0** | **61.5** | **20.5** | **72.4** | 59.5 | 53.0 |

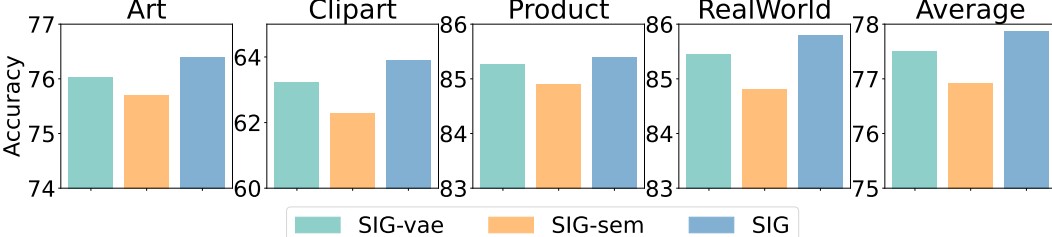

Figure 4: Ablation study on the Office-Home dataset. we explore the impact of different loss terms.

most challenging dataset, DomainNet, which contains more classes and more complex domain shifts. Results in Table 4 show the significant performance of the proposed SIG method, which provides 3.3% averaged promotion, although the performance in the task of Sketch is slightly lower than that of DRT+ST. Compared with iMSDA, our SIG overpasses by a large margin under a more general data generation process.

**Ablation Study:** To evaluate the effectiveness of individual loss terms, we also devise the two model variants. 1) **SIG-sem**: remove the class-aware alignment loss. 2) **SIG-vae**: remove the reconstruction loss and the KL divergence loss. Experiment results on the Office-Home dataset are shown in Figure 4. We can find that the class-aware alignment loss plays an important role in the model performance, reflecting that the class-aware alignment can mitigate the influence of target shift. We also discover that incorporating the reconstruction and KL divergence has a positive impact on the overall performance of the model, which shows the necessity of modeling the marginal distributions.

## 6  Conclusion

This paper presents a general data generation process for multi-source domain adaptation, which coincides with real-world scenarios. Based on this data generation process, we prove that the changing latent variables are subspace identifiable, which provides a novel solution for disentangled representation. Compared with the existing methods, the proposed subspace identification theory requires fewer auxiliary variables and frees the model from the monotonic transformation of latent variables, making it possible to apply the proposed method to real-world data. Experiment results on several mainstream benchmark datasets further evaluate the effectiveness of the proposed subspace identification guaranteed model. In summary, this paper takes a meaningful step for causal representation learning. **Broader Impacts:** SIG disentangles the latent variables to create a model that is more transparent, thereby aiding in the reduction of bias and the promotion of fairness. **Limitation:** However, the proposed subspace identification still requires several assumptions that might not be met in real-world scenarios. Therefore, how further to relax the assumptions, i.e., conditional independent assumption, would be an interesting future direction.

# 7 Acknowledgements

We are very grateful to the anonymous reviewers for their help in improving the paper. This research was supported in part by the National Key R&D Program of China (2021ZD0111501), the National Science Fund for Excellent Young Scholars (62122022), Natural Science Foundation of China (61876043, 61976052), the major key project of PCL (PCL2021A12). This project is also partially supported by NSF Grant 2229881, the National Institutes of Health (NIH) under Contract R01HL159805, a grant from Apple Inc., a grant from KDDI Research Inc., and generous gifts from Salesforce Inc., Microsoft Research, and Amazon Research.

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
