# OpenReview forum: "Subspace Identification for Multi-Source Domain Adaptation"
_NeurIPS.cc/2023/Conference — NeurIPS 2023 spotlight_

### Official Review · Reviewer_bTZb · 2023-06-16

**Soundness:** 3 good
**Presentation:** 3 good
**Contribution:** 3 good
**Rating:** 7
**Confidence:** 4

**Summary:**

This paper considered multiple-source domain adaptation problems in the context of causal representation learning. Specifically, this paper considered quite general scenarios in the domain adaptation by figuring four different representations.

- Domain specific and label irrelevant
- Domain specific and label relevant
- Domain invariant and label irrelevant
- Domain invariant and label relevant

Notably, this paper derived theoretical assumptions via illustrating when we could identify these four representations (or subspace). A VAE based approach in deep learning is further proposed to identify the subspace. Empirical results are further validated in synthetic data, benchmark domain adaptation dataset (including challenging DomainNET dataset)


**Strengths:**

Overall this reviewer feels that this paper has a solid contribution in the theoretical aspects in causal based domain adaptation. Extensive experiments further validated the practical utility.

- **Significance & Originality**
This paper proposes a very solid theoretical justification in the identifiability issue in multi-source da, which is a fundamental problem. More importantly, this paper studied the general setting (or most difficult settings) in multi-source da. Besides, theoretical assumptions are proposed, which is generally non-trivial.

- **Quality**
The theoretical assumptions and proof seem valid and reasonable for me. The proposed practical method is also reasonable. Empirical results in toy data and real-data demonstrated that the proposed method could better identify different representations.

- **Clarity**
In general, this paper is clearly written and well-explained. Since this paper is a bit theoretical, there are still several theoretical points that need to be clarified (see the weakness part).


**Weaknesses:**

Since this paper covers several fundamental proofs, this reviewer feels uncertain about several points within the paper.


1. Theorem 1 about theoretical assumptions. In general, I could follow the proof, where intuitions within these assumptions could be better explained. I just take theorem 1 as an example.

- About conditional independence. If I understand correctly, this implies, the representation variable $P(Z|U) = \prod P(z_i|U)$ ?  (sort of disentanglement or mean-field approximation assumption)? If we write it as a log probability term, it will be $\log P(Z|U) = \sum \log P(Z_i|U)$, which is consistent with your paper…
- About linear independence. This assumption seems to illustrate that the gradient of the probability density over each component should be linearly independent? That implies if we use a gradient based approach to learn each component, each component will converge to a distinct direction. Thus we could possibly identify different independent components?
- I have a general question in theory 1 (for example). Is it possible to extend this assumption into the blockwise vectors? i.e, $P(Z|U) = P(Z_1|U)P(Z_2|U)P(Z_3|U)P(Z_4|U)$, where $Z_1$ is a subspace vector rather than a scalar.

2. About explanations in toy data. It seems that the proposed method works quite significantly in toy data. However the experimental details (such as the data generation) could be better clarified. For example, is there some sort of spurious correlation?

3. In Equation (1) Line 3, I would think this holds when we have figure 1 by considering the conditional independence.

4. I like the example in explaining different subspace in line 82-92. It could be better to consider this example into more real-world scenarios such as drug discovery and health. For example, data collected from different hospitals may create different $Z_1$, this could be helpful to better understand the importance of considering a general scenario.


**Questions:**

See comments in the weakness part.

---

> ### Author Rebuttal · Authors · 2023-08-08
>
> We are very grateful for your valuable comments, helpful suggestions, and encouragement. Below please see our point-to-point responses to your comments and suggestions.
>
> >**Q1**: Theorem 1 about theoretical assumptions. In general, I could follow the proof, where intuitions within these assumptions could be better explained. I just take theorem 1 as an example.
>
> >1) About conditional independence. If I understand correctly, this implies, the representation variable $P(z|u)=\prod(z_i|u)$
>  ? (sort of disentanglement or mean-field approximation assumption)? If we write it as a log probability term, it will be $\log P(z|u)=\sum P(z_i|u)$, which is consistent with your paper…
> >2) About linear independence. This assumption seems to illustrate that the gradient of the probability density over each component should be linearly independent. That implies if we use a gradient-based approach to learn each component, each component will converge in a distinct direction. Thus we could possibly identify different independent components.
> >3) I have a general question in theory 1 (for example). Is it possible to extend this assumption into the blockwise vectors? i.e, $P(z|u)=P(z_1|u)P(z_2|u)P(z_3|u)P(z_4|u)$, where $z_1$ is a subspace vector rather than a scalar.
>
> **A1**: I would like to express my sincere gratitude for your meticulous review. We answer these questions point by point.
>
> 1) s for conditional independence, I think that the understanding of the reviewer, where the representation variables follow the mean-field approximation assumption, is correct. The intuition of conditional independence is that the changing factors are independent of each other. For example, in the image classification scenario, the light directions of images and the resolution of images change independently.
> 2) I think there is some misunderstanding between the derivative of latent variables and the derivative of model parameters. On one hand, the linear independence assumption implies that the $w(z,u_j)-w(z,u_0)$ are linearly independent, and each element in $w(z,u_j)=(\frac{\partial q_1(z_1,u)}{\partial z_1},\cdots, \frac{\partial q_i(z_i,u)}{\partial z_i},\cdots \frac{\partial q_{n_s}(z_{n_s},u)}{\partial z_{n_s}})$ denotes the first-order derivative of the ground truth log probability density $q_i(z_i,u)$ with respect to ground-truth latent variables $z_i$. On the other hand, the gradient of the gradient-based approaches is calculated by the derivative of the loss with respect to the parameters of the models. These two derivatives are different.
> 3) Existing disentanglement methods based on nonlinear ICA usually assume that each dimension of latent variables is independent of each other. However, there are some cases that the independent assumption can not be met. So disentanglement under dependent latent variables ($p(z|u)=P(z_1|u)P(z_2|u)P(z_3|u)P(z_4|u)$ is a special case of dependent latent variables) is a future direction. In this scenario, homogeneous linear equations are hard to be developed since the latent variables are not independent, but the property of independence of latent variables given their Markov Blanket can be used to develop the homogeneous linear equations.
>
> >**Q2**: About explanations in toy data. It seems that the proposed method works quite significantly in toy data. However, the experimental details (such as the data generation) could be better clarified. For example, is there some sort of spurious correlation?
>
> **A2**: In light of your valuable suggestions, we have reorganized the causal generation process. In detail, we generate the simulation data for binary classification with 8 domains, which follows the data generation process as shown in Figure 2, which includes two types of latent variables, i.e., domain-specific latent variables $z_s$ and domain-invariant latent variables $z_c$. To better evaluate the subspace identification results, we let the label distribution is the same across different domains, so there is no spurious correlation. The domain-specific latent variables $z_s$ are sampled from 8 different mixtures of Gaussians and $z_c$ are sampled from a factorized Gaussian distribution. As for the nonlinear generation process, we let the observed variables be generated via an MLP with the Tanh activation function.
>
>
> >**Q3**: In Equation (1) Line 3, I would think this holds when we have Figure 1 by considering conditional independence.
>
> **A3**: Thank you for your reminder. We totally agree with you, and leverage the property of the causal generation process when deriving Equation (1). In light of your suggestion, we have provided a clear explanation of how to derive Equation (1). In detail, the derivation in Equation (1) can be separated into three steps.
>
> 1) We introduce the latent variables $z_1,z_2,z_3$, and $z_4$, which have mentioned in Section 2.1.
> 2) We factorize the joint distribution in Equation (1) into $p_{x,z_1,z_2,z_3,z_4|y,u_{\mathcal{T}}}$ and $p_{y|u_{\mathcal{T}}}$ with the help of Bayes Rule.
> 3) We further use Bayes Rule to factorize $p_{x,z_1,z_2,z_3,z_4|y,u_{\mathcal{T}}}$. Since $x$ is conditional independent of $u, y$ given $z_1,z_2,z_3,z_4$, we can obtain $p_{x|z_1,z_2,z_3,z_4}$.
>
> >**Q4**: ...It could be better to consider this example in more real-world scenarios such as drug discovery and health. For example, ....
>
> **A4**: Thank you for your affirmation. In the scenario of health, we try to provide an example of malaria detection, $z_1$ denotes the domain-specific information, for example, the children's hospital and the contagious hospital; $z_2$ denotes the latent variables that can be age relevant to domains and labels. For example, infants are more likely to suffer from malaria, and the age of patients are also influenced by the different hospital; $z_3$ denotes the symptoms of patients; And $z_4$ denotes the label-irrelevant latent variables like gender.

---

> > ### Comment · Reviewer_bTZb · 2023-08-14
> >
> > Thanks for your rebuttal. I think it has addressed my confusions on the theoretical parts. I think it could be useful to include a short discussion about the intuition on different assumptions.  After further checking others' reviews, I recommend acceptance.

---

> > > ### Author Response · Authors · 2023-08-15
> > >
> > > We are glad to address your confusion of theories and thank you for the valuable suggestions. We will provide a short discussion about the intuition of the assumptions.
> > >
> > > With best wishes,
> > >
> > > Authors of submission #4882

---

### Official Review · Reviewer_pec6 · 2023-07-07

**Soundness:** 3 good
**Presentation:** 3 good
**Contribution:** 3 good
**Rating:** 8
**Confidence:** 4

**Summary:**

This paper studies the problem of multi source domain adaptation where we have access to multiple labeled source domains and unlabeled target domain. Authors consider a novel data generation process by modeling them through 4 new latent variables (i.e. combinations of domain specific or domain invariant and label relevant/ label irrelevant). This is done to relax the assumptions made by previous theoretical analyses. The authors provide sufficient theoretical analysis in this context by providing subspace identification guarantees and using that to improve the results. They conduct multiple experiments include synthetic experiments and the results are satisfactory.


**Strengths:**

*  The paper is well-presented.
* The paper provides new framework through their modeling of data generation process by introducing the 4 latent variables and then considering them to influences from the domain index and the target labels.
* Their theoretical analysis is convincing and experiments are impressive.


**Weaknesses:**

* This framework is somewhat similar to an earlier work on domain generalization(https://arxiv.org/pdf/2102.11436.pdf) . I agree that your data generation process further expands on their structural causal model but noting the similarities it should be cited and expand on the differences
* Importantly, since we model label relevant attributes separately, should we still make the assumption that source and target contain the same labels.
* My question specifically is can you comment on if this framework naturally support open-set/partial MSDAs.


**Questions:**

Please comment on the weakness I raised above and if you could provide some results even on one dataset where there is either a lesser number of classes in target than in source (https://arxiv.org/abs/1808.04205)  or like in other open set DA problems that would be great


**Limitations:**

Authors have adequately addressed the limitations.

---

> ### Author Rebuttal · Authors · 2023-08-08
>
> We are very grateful for your valuable comments, helpful suggestions, and encouragement. Below please see our point-to-point responses to your comments and suggestions.
>
> >**Q1**: This framework is somewhat similar to earlier work on domain generalization(https://arxiv.org/pdf/2102.11436.pdf). I agree that your data generation process further expands on their structural causal model but noting the similarities it should be cited and expand on the differences
>
> **A1**: We sincerely appreciate your recommendation on these methods, which help clarify the difference between our contributions and other works. In light of your suggestions, we have included the discussions and comparisons in the revised related work. Several methods [1][2][3] employ the prior causal structure to address the distribution shift challenges. Although these methods leverage causal graphs to investigate how the distributions change, the proposed method employs a different causal generation process, which considers different types of domain shifts and is more general. Moreover, our method considers different types of latent variables and provides subspace identification guarantees.
>
> [1] Model-Based Domain Generalization Alexander Robey, George J. Pappas, Hamed Hassani, NeurIPS2021
>
> [2] Learning Disentangled Semantic Representation for Domain Adaptation, Ruichu Cai, Zijian Li, Pengfei Wei, Jie Qiao, Kun Zhang, Zhifeng Hao, IJCAI2019
>
> [3] Domain Adaptation under Target and Conditional Shift, Kun Zhang, Bernhard Sch¨olkopf, Krikamol Muandet, Zhikun Wang, ICML2013
>
> >**Q2, Q3, Q4:** Importantly, since we model labels relevant attributes separately, should we still make the assumption that source and target contain the same labels?
>  My question specifically is can you comment on if this framework naturally supports open-set/partial MSDAs.
>  Please comment on the weakness I raised above and if you could provide some results even on one dataset where there is either a lesser number of classes in target than in source (https://arxiv.org/abs/1808.04205) or like in other open set DA problems that would be great
>
> A: Thanks for your treasure suggestions. We would like to highlight that the proposed SIG is a general framework for multi-source domain adaptation. In this paper, we employ the standard multi-source domain adaptation setting to evaluate our method, where the label space of the source and target domain is assumed to be the same. Generally speaking, our method can extend to other scenarios, such as open-set MSDAs and partial MSDAs. As you mentioned, it is because we model labels relevant attributes separately, and hence allow the setting where source and target domains contain different labels.
>
> For example, in partial MSDAs[4], the source label space is a superset of the target label space. One of the challenges of partial MSDAs is to mitigate the influence of the source labeled data in outlier label space. In light of your suggestion, we have extended our method to the partial domain MSDAs setting, following the paradigm of PADA[4] and using the pseudo label to estimate the target label space. Specifically, we apply the class-aware conditional alignment on the target label space, where the confidence of the source-specific label is low. Moreover, we have provided the experimental results of the partial MSDAs in the following table, where five labels are removed in the target domain to satisfy the partial MSDA setting. We find that our SIG model achieves superior performance.
>
> |       |   Art | Clipart | Product | RealWorld | Average |
> |-------|-------|---------|---------|-----------|---------|
> | SIG |  76.0 |    62.7 |    85.8 |      85.6 |    77.5 |
> | PSDA  |  75.5 |    60.4 |    83.3 |      83.8 |    75.7 |
>
> [4] Partial Adversarial Domain Adaptation, Zhangjie Cao, Lijia Ma, Mingsheng Long, Jianmin Wang,ECCV2018

---

> > ### Comment · Reviewer_pec6 · 2023-08-14
> > **Answers my questions.**
> >
> > Dear Authors,
> > Thank you for taking the time and running some more experiments. I appreciate the effort and I note the improvements in MSDA settings too. I am happy to increase my score.

---

> > > ### Author Response · Authors · 2023-08-14
> > >
> > > We are very happy that you found the response well addressed your concerns. Thank you once again for your valuable comments and suggestions and for championing our submission.
> > >
> > > With best wishes,
> > >
> > > Authors of submission #4882

---

### Official Review · Reviewer_nEWG · 2023-07-07

**Soundness:** 3 good
**Presentation:** 3 good
**Contribution:** 3 good
**Rating:** 7
**Confidence:** 3

**Summary:**

In this paper, the authors investigate the problem of multi-source domain adaptation. In detail, the authors first devise a causal generation process, which consider the domain-specific and label-irrelevant, domain-specific and label-relevant, domain-invariant and label-invariant, domain-invariant and label-irrelevant variables. Based on the aforementioned generation process, the authors raise the identification guarantees for the latent variables. The authors also evaluate the proposed methods on several benchmarks and achieve good performance.

**Strengths:**

1.	The authors provide an interesting perspective for multi-source domain adaptation, which uses nonlinear ICA to identify the latent variables. Compared with the existing methods, the proposed method relaxes the assumptions.
2.	The authors evaluate the proposed methods on several datasets and achieve the ideal results.


**Weaknesses:**

1.	According to the theoretical results, the authors give three assumptions which is similar to those of nonlinear ICA literature. Why are these existing assumptions applicable in real-world scenarios? Moreover, according to the subspace identification results, the dimension of latent variables is small and restricted by the number of domains, which is hard to be met in practice.
2.	For the real-world scenario, the number of ground-truth latent variables is unknown, is it reasonable to assume that their number is according to the observed domains?
3.	The authors equip a cross-attention module to the ResNet101 for the DomainNet dataset, it is suggested the authors provide the experiment results of other datasets, e.g., Office-home, with the proposed cross-attention module.

Minors:
There are some typos in this paper. For example, the illustrations of Lemma2 in the main and appendix are inconsistent.

**Questions:**

NO

**Limitations:**

NON

---

> ### Author Rebuttal · Authors · 2023-08-08
>
> We would like to express our deep gratitude for the valuable feedback and helpful suggestions provided on our paper, as well as for the time you devoted to reviewing it. Below, we have addressed each of your comments and suggestions.
>
> >**Q1**: According to the theoretical results, the authors give three assumptions that are similar to those of nonlinear ICA literature. Why are these existing assumptions applicable in real-world scenarios? Moreover, according to the subspace identification results, the dimension of latent variables is small and restricted by the number of domains, which is hard to be met in practice.
>
> **A1**: Thanks for your great question. The assumptions used in this paper are reasonable and mild. We explain them one by one.
>
> 1) Assumption 1 implies that $p(z|u)$ is smooth and positive, i.e., the domains are changing continuously, for example, the light directions of images are changing continuously.
>
> 2) Assumption 2 means that $p(z|u)=\prod p(z_i|u)$, i.e., changing factors are independent of each other, for example, the light directions of images and the resolution of images are independent.
>
> 3) Assumption 3 indicates that the latent variables can be identified when the number of domains is sufficient. Since it is easy to access the data with different distributions. Hence, these assumptions are applicable in real-world scenarios.
>
> In practice, the changes between different domains can be described via a small dimension, for example, the change of light directions of images can be described via a one-dimension variable. It is a common minimal change assumption in causality[1]. Therefore, the small dimension of latent variables can be met in practice. We will add a discussion about the reasonableness of the assumptions in the final version.
>
> [1] Domain Adaptation with Invariant Representation Learning: What Transformations to Learn? Petar Stojanov, Zijian Li, Mingming Gong, Ruichu Cai, Jaime Carbonell, Kun Zhang, NeurIPS2021
>
>
> >**Q2**: For the real-world scenario, the number of ground-truth latent variables is unknown, is it reasonable to assume that their number is according to the observed domains?
>
> **A2**: Thank you very much for the profound question to help us clarify the implementation of our work. Due to the minimal change assumption mentioned above, the dimension of the changing variables is usually small. Besides, since the latent variables $z_2$ are caused by the domains $u$ and $y$, the dimension of $z_2$ is restricted by $|u|\times|y|-1$, which is large enough and hence is easy to satisfy in real-world scenarios. For example, the DomainNet dataset contains 6 domains and 345 categories of objects, indicating that the dimension of latent variables can be 2070. Therefore, it is reasonable to assume that their number is according to the observed domains.
>
>
> >**Q3**: The authors equip a cross-attention module to the ResNet101 for the DomainNet dataset, it is suggested the authors provide the experiment results of other datasets, e.g., Office-home, with the proposed cross-attention module.
>
> **A3**: We are grateful for your careful review and constructive suggestion to improve the completeness of our experiments. In light of your suggestion, we have provided the experiment results on the Office-home dataset with the cross-attention module, named SIG+CA, which is shown in the table below. We find that the SIG+CA also achieves comparative results, showing the effectiveness of the cross-attention module.
>
> |       |   Art | Clipart | Product | RealWorld | Average |
> |-------|-------|---------|---------|-----------|---------|
> | Source Only   |  64.5 |    52.3 |  77.6 |  80.7 |  68.8 |
> | DANN   |  64.2 |  58.0 |  76.4 |   78.8 |  69.3 |
> | DAN   |  68.2 |  57.9 |  78.4 |   81.9 |   71.6 |
> | DCTN  |  66.9 |  61.8 |  79.2 |   77.7 |  71.4 |
> | MFSAN   | 72.1 |  62.0 |  80.3 |  81.8 |  74.1 |
> | WADN  |  75.2 |    61.0 |  83.5 |      84.4 |    76.1 |
> | iMSDA   |  75.4 |    61.4 |    83.5 |      84.4 |    76.2 |
> | SIG   |  76.4 |    63.9 |    85.4 |      85.8 |    77.8 |
> | SIG+CA |  75.8 |    61.8 |    84.0 |      84.7 |    76.6 |
>
> >**Q4**: There are some typos in this paper. For example, the illustrations of Lemma2 in the main and appendix are inconsistent.
>
> **A4**: Thanks for noticing it and kindly letting us know! We have read the paper carefully and corrected the typos.

---

> > ### Comment · Reviewer_nEWG · 2023-08-17
> >
> > Greatly appreciate the responses from the reviewers.

---

### Official Review · Reviewer_Yx4K · 2023-07-09

**Soundness:** 3 good
**Presentation:** 2 fair
**Contribution:** 3 good
**Rating:** 7
**Confidence:** 4

**Summary:**

This paper proposes a novel framework to tackle the problem of multi-source unsupervised domain adaptation. Existing methods often have stringent requirements, such as a large number of domains and invariant label distributions. To address these limitations, this paper proposes a subspace identification theory that disentangles domain-invariant and domain-specific variables under less restrictive constraints. The proposed Subspace Identification Guarantee (SIG) model is based on variational inference and additionally incorporates class-aware conditional alignment to handle domain shifts. Experimental results demonstrate that the SIG model outperforms existing SOTA techniques on four benchmark datasets (OfficeHome, ImageCLEF, PACS, DomainNet), showcasing its effectiveness in real-world applications.

**Strengths:**

- The paper defines three major drawbacks of [a]  – invariant label distribution, requiring a large number of domains and monotonic transformation between latent variables. These assumptions are relaxed by the *Subspace Identification Guarantee* (SIG) model proposed in this paper. Most interestingly, while Kong et al. claimed that for $n$ dimensional latent space, $2n+1$ domains would be necessary, authors here have shown that “n+1” domains are sufficient.
- The core ideas on subspace identifiability are theoretically grounded. The paper defines a new data generative process based on a 4-way split of the latent space and builds upon the theoretical results to define a new multi-source unsupervised domain adaptation framework using a VAE-based architecture.
- Results are presented on 4 benchmark datasets, with the proposed model outperforming all other SOTA approaches.

[a] Kong et. al., Partial Identifiability for Domain Adaptation, ICML 2022

**Weaknesses:**

- In the data generation process, the paper introduces 4 latent variables from the product of two sets {domain-specific, domain-invariant} $\times$ {label-relevant, label-irrelevant}. I am unsure what the intuitive reasoning behind the "domain-invariant label-irrelevant" latent variable $\mathbf{z}_4$ is.
- There are no individual constraints on $\mathbf{z}_1$ and $\mathbf{z}_4$ during the learning process. Are they identifiable individually?
- While the paper claims that the proposed SIG model can handle all three types of shifts: covariate, target, and conditional, experiments focus primarily on covariate shift (based on the datasets used). It’s unclear how the datasets, which contain mostly covariate shifts, were used for studying the two other shifts.
- The clarity could be improved, especially when introducing the SIG model. For example, in Section 4.2, $\hat{\mathbf{z}}_{3,\mathcal{S}}^{(i)}$ is denoted as "..latent variables of i-th class from source". How is this estimated?

**Questions:**

Please see weaknesses. Also, the derivation of the ELBO in Equation 5 should be provided to enhance the completeness of the paper.

**Limitations:**

The authors discuss the limitations adequately.

---

> ### Author Rebuttal · Authors · 2023-08-08
>
> We highly appreciate the valuable commands and helpful suggestions on our paper and the time dedicated to reviewing it. Below please see our point-to-point responses to your comments and suggestions.
>
> >**Q1**: I am unsure what is the intuitive reasoning behind the "domain-invariant label-irrelevant" latent variable $z_4$ is.
>
> **A1**: Thanks for your valuable questions which have improved the readability of our paper. We have provided the intuitive reasoning behind the domain-invariant and label-irrelevant latent variables $z_4$ and will include it in the final version. Although $z_4$ is not unrelated to domains or labels, it is an important component of the causal generation process. Moreover, reasoning $z_4$ with the block-wise identification guarantee can disentangle $z_4$ with other latent variables i.e., $z_1, z_2, z_3$, which makes it convenient to precisely identify other latent variables with subspace identification guarantee.
>
> >**Q2**: There are no individual constraints on $z_1$ and $z_4$  during the learning process. Are they identifiable individually?
>
> **A2**: We deeply value your careful reviews for the preciseness of our work. Actually, $z_1$ and $z_4$ are not identified individually. According to ELBO as the equation below, we can employ $\textcolor{blue}{\ln p_{{u}|{z}}({{u}|{z}})= \ln p_{{u}|{z}} ({u}| z_1,z_2,z_3)}$ as a constraint on $z_1$. It indicates that $z_1$ is identified with domain labels. Moreover, we employ $\textcolor{green}{\ln{p_{x|{z}}}({x}|{z})=\ln{p_{x|{z}}}(x|{z}_1,{z}_2,{z}_3,{z}_4)}$ as a constraint on $z_4$, indicating that $z_4$ is with block-wise identification.
>
> $ELBO = E_{q_{z|x}(z|x)}[\textcolor{green}{ln p_{x|z}(x|z)}+ln p_{y|u,z_{2},z_{3}}(y|u,z_{2},z_{3})+\textcolor{blue}{\ln p_{u|z}(u|z)}]-D_{KL}(q_{z|x}(z|x)||p_{z}(z))$
>
> In the implementation, since the reconstruction of $u$ is not the optimization goal and might bring insignificant complexity of optimization, we remove the reconstruction of $u$ as mentioned in Line 200 of Page 5.
>
> In light of your suggestions, we further conduct an experiment to evaluate the effectiveness of the reconstruction of $u$. Specifically, we add a domain classifier to our SIG, named SIG+D. Experimental results are shown in the following table. According to the experiment results, we can find that the SIG+D also achieves a comparative performance and the standard SIG can achieve an ideal performance without introducing extra complexity.
>
> ||Art|Clipart|Product|RealWorld|Average|
> |-|-|-|-|-|-|
> |SIG|76.4|63.9|85.4|85.8|77.8|
> |SIG+D|76.1|63.2|85.6|85.8|77.7|
>
> >**Q3**: It’s unclear how the datasets, which contain mostly covariate shifts, were used for studying the two other shifts.
>
> **A3**: Thanks a lot for this question. We have tried our best to include the most mainstream benchmark datasets for multi-source domain adaptation.  While most of them follow the covariate shift, some of them consist of other types of domain shift like conditional shift and target shift. For example, in office-home datasets, pictures with monitor, computer, and laptop are tagged with the same label, meaning that the covariate shift assumption ($p_{\mathcal{S}}(y|x)=p_{\mathcal{T}}(y|x)$) does not always hold, but conditional shift ($p_{\mathcal{S}}(x|y) \neq p_{\mathcal{T}}(x|y)$) holds. Moreover, $p(y)$ also slightly changes across domains. Therefore, these datasets contain a mixture of three types of shifts instead of covariate shifts.
>
> In light of your suggestion, we design an experimental setting where the target shift changes across domains. Experiment results on SIG, iMSDA, and WADN are shown in the table below. We find that our method achieves the best performance, evaluating the effectiveness of the target shift scenario.
>
> ||Art|Clipart|Product|RealWorld|Average|
> |-|-|-|-|-|-|
> |SIG|75.4|62.0|84.9|85.2|77.8|
> |iMSDA|73.2|57.2|82.6|84.2|74.3|
> |WADN|74.8|60.6|84.1|84.2|75.9|
>
> >**Q4**:  ...For example, in Section 4.2, $\hat{z}_{3,\mathcal{S}}^{(i)}$ is denoted as "..latent variables of $i$-th class from source". How is this estimated?
>
> **A4**: We sincerely appreciate your suggestion of our paper for better clarity. We estimate $\hat{z}_{3,\mathcal{S}}^{(i)}$ via a moving average method as follows:
>
> 1) For each class, we initialize $\hat{z}_{3,\mathcal{S}}^{(i,0)}$ in the first training step.
> 2) In the $\tau$ training step, we randomly sample a batch of sample $(x_k, y_k)_{k=1}^{B}$ with the size of $B$ from source domain $\mathcal{S}$.
> 3) We estimate $z_3$ of each sample via the pre-trained backbone networks and the bottleneck networks, so we obtain $(z_{3,k}, y_k)_{k=1}^{B}$.
> 4) Given the $i$-th class, we calculate $c_{3,\mathcal{S}}^{(i)}=\frac{1}{B_i} \sum \limits_{(z_{3,k},y_k=i)} z_{3,k}$, where $B_i$ is the sample number of the $i$-th class.
> 5) Sequentially, we update the estimated variables via $\hat{z}_{3,\mathcal{S}}^{(i,\tau)}=\gamma c_{3,\mathcal{S}}^{(i)} + (1-\gamma) \hat{z}_{3,\mathcal{S}}^{(i,\tau-1)}$ where $\gamma$ is a hyper-parameter.  *(We also provided a PDF version of Q4 due to some issues related to the Markdown formatting and LaTeX equations in OpenReview)*
>
> In light of your valuable advice, we have added an algorithm in Appendix to introduce how to estimate $\hat{z}_{3,\mathcal{S}}^{(i)}$ and will include it in the final version.
>
> >**Q5**: The derivation of the ELBO in Equation 5 should be provided to enhance the completeness of the paper.
>
> **A5**: Thank you very much for this suggestion to improve the completeness of our paper. In light of your suggestion, we have provided the derivation of the ELBO in the Appendix, which is shown as follows.
>
> ${\quad}\ln p(x, y, u) = \ln \frac{p(x, y, u, z)}{p(z|x, y, u)}=\ln \frac{p(x|z)p(u,y,z)}{p(z|x, y, u)}=\ln \frac{p(x|z)p(y|u,z_2,z_3)p(u|z)p(z)}{p(z|x, y, u)}$
>
> $=E_{q(z|x)}\ln\frac{p(x|z)p(y|u,z_2,z_3)p(u|z)p(z)}{q(z|x)} + D_{KL}(q(z|x)||p(z|x,y,u))$
>
> $\geq E_{q(z|x)}\ln p(x|z) + E_{q(z|x)} \ln p(y|u,z_2,z_3) + E_{q(z|x)} \ln p(u|z) - D_{KL}(q(z|x)||p(z)) = ELBO$

---

### Author Rebuttal · Authors · 2023-08-08

Dear Reviewers Yx4K, nEWG, pec6, and bTZb:

Thanks for the thoughtful and constructive review. It is encouraging that the reviewers think SIG is novel (Reviewer Yx4K and pec6), interesting (Reviewer Yx4K and nEWG), and solid ( Reviewer bTZb). We here provide a general response to summarize the modifications of the paper.

- To Reviewer Yx4K, we have clarified the intuition of the "domain-invariant and label-irrelevant" latent variables $z_4$.
- To Reviewer Yx4K, we have explained the identifiability of $z_1$ and $z_4$, and provided experimental results according to your suggestions.
- To Reviewer Yx4K, we have clarified how the datasets are used to study the conditional shift and target shift. And we also added experimental results in light of your suggestions.
- To Reviewer Yx4K, we have added the derivation of the ELBO in Equation (5).
- To Reviewer nEWG, we have explained the reasonableness of the assumptions.
- To Reviewer nEWG, we have explained the assumptions and the reasonableness for the number of latent variables.
- To Reviewer nEWG, we have added the experimental results of our SIG with the cross-attention module of the OfficeHome dataset.
- To Reviewer nEWG, we have read the paper carefully and corrected the typos.
- To Reviewer pec6, we have clarified the difference between our contributions and other works as your recommended.
- To Reviewer pec6, we have discussed the assumption of the same label space of source and target domains.
- To Reviewer pec6, we have commented on the extension to open-set/partial MSDA of our method and added experimental results of partial MSDA.
 - To Reviewer bTZb, we have clarified the intuition of assumptions.
 - To Reviewer bTZb, we have discussed the conditional independence of Equation (1).
 - To Reviewer bTZb, we have provided an example in the health scenario.

Thanks again for your time dedicated to carefully reviewing this paper. We hope that our response properly addresses your concerns.

With best regards,

Authors of submission 4882

---

### Decision · Program_Chairs · 2023-09-21

**Decision:**

Accept (spotlight)

**Comment:**

This paper received strong reviews with one strong accept and three accept recommendations. The paper addresses the problem of multi-source unsupervised domain adaptation and focuses on reducing the assumptions required in prior methods (invariant label distributions, large number of domains etc). The proposed method of subspace identification guarantee (SIG) was found to be novel, interesting, and theoretically grounded by the reviewers. Further, the method achieved state of the art performance across four standard datasets. The rebuttal/discussion between the authors and reviewers was productive with many of the reviewers questions being answered and positively impacting their final recommendations. Given the strength of this work and unique direction it proposes, the AC recommends this work be highlighted at the conference.